# DySarl: Dynamic Structure-Aware Representation Learning for Multimodal Knowledge Graph Reasoning

## ABSTRACT

Multimodal knowledge graph (MKG) reasoning has attracted significant attention since impressive performance has been achieved by adding multimodal auxiliary information (i.e., texts and images) to the entities of traditional KGs. However, existing studies heavily rely on path-based methods for learning structural modality, failing to capture the complex structural interactions among multimodal entities beyond the reasoning path. In addition, existing studies have largely ignored the dynamic impact of different multimodal features on different decision facts for reasoning, which utilize asymmetric coattention to independently learn the static interplay between different modalities without dynamically joining the reasoning process. We propose a novel **Dy**namic **S**tructure-**a**ware **r**epresentation **l**earning method, namely **DySarl**, to overcome this problem and significantly improve the MKG reasoning performance. Specifically, we devise a dual-space multihop structural learning module in DySarl, aggregating the multihop structural features of multimodal entities via a novel message-passing mechanism. It integrates the message paradigms in Euclidean and hyperbolic spaces, effectively preserving the neighborhood information beyond the limited multimodal query paths. Furthermore, DySarl has an interactive symmetric attention module to explicitly learn the dynamic impacts of unimodal attention senders and multimodal attention targets on decision facts through a newly designed symmetric attention component and fact-specific gated attention unit, equipping DySarl with the dynamic associations between the multimodal feature learning and later reasoning. Extensive experiments show that DySarl achieves significantly improved reasoning performance on two public MKG datasets compared with that of the state-of-the-art baselines. Source codes are available at https://anonymous.4open.science/r/DySarl.

## CCS CONCEPTS

• **Computing methodologies → Knowledge representation and reasoning**.

## KEYWORDS

Multimodal knowledge graph, graph convolutional network, cross-modal fusion

## 1 INTRODUCTION

Reasoning over multimodal knowledge graphs (MKGs) has attracted significant attention due to its greater conformity to real-world complex scenarios, such as multimodal retrieval and social media analysis. An MKG is essentially a multirelational graph composed of multimodal nodes (entities). It contains not only the structural triples (*subject*, *relation*, *object*) of traditional KGs but also rich multimodal auxiliary information (e.g., texts and images).

Two fundamental issues need to be well addressed to perform good MKG reasoning. 1) At the graph level, how can the complex structural features among multimodal entities (also known as the structural modality) be effectively learned? 2) At the entity level, how can the auxiliary features of different modalities be efficiently fused? We argue that existing studies fail to properly learn the very complex graph-level structural modality (i.e., multimodal multihop structures) and entity-level cross-modal attentive dynamics, hindering performance improvements in MKG reasoning.

To effectively learn the structural modality, early methods [4, 21, 32] merely focus on limited one-hop information. Recently, a series of multihop path learning methods [30, 39, 40] have been applied to MKGs and achieved state-of-the-art reasoning performance. However, these methods fail to capture complex structural interactions beyond query paths with a very limited number of hops. As shown in Figure 1(a), in an MKG consisting of multimodal entities, the query decision fact is denoted as (*Joe Biden*, *Live At*, ?), and "White House" is the ground-truth answer. Then, the traditional multihop path learning approaches merely focus on the orange path "*Joe Biden* $\overset{President}{\longleftarrow}$ *USA* $\overset{Capital}{\longrightarrow}$ *Washington* $\overset{LocatedIn}{\longleftarrow}$ *White House*" to perform reasoning. Nevertheless, more structural information outside the query path valuable for factual reasoning is ignored. Taking the blue relational facts presented in Figure 1(a) as examples, the structural neighborhoods beyond multihop paths can provide more useful features for the query (*Joe Biden*, *Live At*, ?), such as its political context (*Joe Biden*, *MemberOf*, *Democratic Party*) and abstract concept (*White House*, *SubclassOf*, *Building*). There exists multihop structural learning methods [13, 15, 20, 28] for unimodal multirelational graphs but they are not able to integrate the complex modalities of entities.

To efficiently fuse entity-level multimodal features, early approaches [9] only integrate the coarse-grained information of different modalities through vector concatenation. Some works [33, 37] used conventional self-attention mechanisms [29] for extracting the fine-grained image features of entities, but these methods neglect the fine-grained attention interactions between various modalities. Coattention mechanisms [14, 38, 40] are later proposed to address this issue and achieved significant performance improvements. However, none of them address the dynamic effects of different modalities on various decision facts when reasoning. As shown in Figure 1(b), in the cross-modal feature fusion stage, the

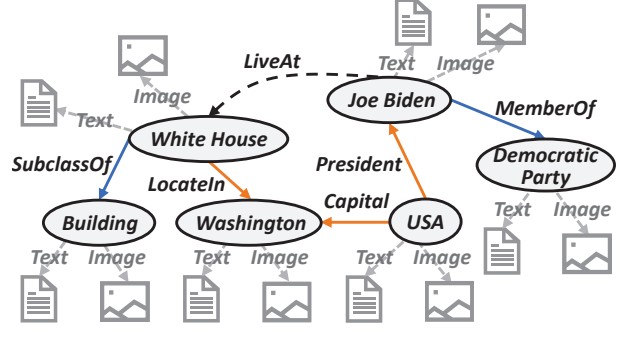

**(a) Illustration of multimodal multihop structures**

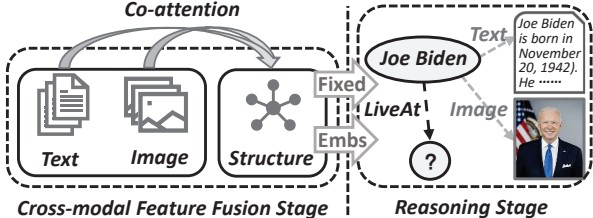

**(b) Illustration of cross-modal attentive dynamics**

**Figure 1: Illustration of the challenges of multimodal multihop structures and cross-modal attentive dynamics.**

previous approaches [4, 40] treat certain modal features (e.g., texts and images) as attention senders; thus, the attention target learning process excludes these modalities and focuses on capturing their effects on the target modal features (e.g., structures). Moreover, in the traditional codec-based architecture, the feature fusion stage (the encoder part) delivers fixed entity embeddings to the reasoning stage (the decoder part). As presented in Figure 1(b), the fixed entity embeddings remain static during the score calculation process in the reasoning stage, consequently limiting the contributions of the diverse multimodal entity-level features to different decision facts in the reasoning stage and resulting in poor reasoning performance.

**Our contributions**. To further fill this research gap, we propose a novel MKG reasoning method, namely **DySarl**, which properly learns the graph-level multimodal multihop structures and entity-level cross-modal attentive dynamics via **Dy**namic **S**tructure-**a**ware **r**epresentation **l**earning. To properly learn the complex multimodal multihop structure, we propose a Dual-space Multihop Structural learning (DMS) module. Specifically, as shown in Figure 2(a), we leverage the multilayer graph neural network (GNN) architecture and carefully devise a new message-passing mechanism to aggregate the multihop structural features among the multimodal entities in MKGs. Furthermore, it integrates the message paradigms in the Euclidean space and hyperbolic space to learn the association-based and hierarchy-based structural features, effectively preserving the complex neighborhood information beyond the limited multimodal query paths.

To capture the cross-modal attentive dynamics during the entity-level feature fusion process, we propose an Interactive Symmetric Attention fusion (ISA) module. As shown in Figure 2(c), a novel attention mechanism is devised to learn the dynamic influences

**Table 1: Summary of the existing MKG reasoning methods.**

| Models \ Types
KGs | Single-hop | Multihop path | Multihop structure |
|---|---|---|---|
| Unimodal KGs | TransE, RotatE
ComplEx, DistMult
ConvTransE, ATTH | DeepPath, MINERVA
FIRE, GaussianPath
RLH, NeuralLP | RGCN, CompGCN
StarE, KBGAT
REGCN, ReTIN |
| Multimodal KGs | IKRL, TransAE
KR-AMD, MKRL
MTRL, OTKGE | MMKGR | **DySarl** |

of different modal entity explicitly features on decision facts, symmetrically and uniformly treating all modal information in MKGs as attention targets for factual inference. Meanwhile, we incorporate a carefully designed learnable fact-specific gated attention unit to establish dynamic associations between multimodal feature learning and later reasoning, learning dynamic weights to the entity features of different decision facts. Hence, the effects of both unimodal attention senders and dynamically learned multimodal attention targets on reasoning can be well captured. This significantly enhances the symmetry of the cross-modal feature fusion process in the ISA module.

Our contributions are summarized as follows.

- We propose a novel framework to address the challenges of graph-level multimodal multihop structures and entity-level cross-modal attentive dynamics in MKG reasoning.
- To aggregate the multihop structures among the multimodal entities, we design a DMS module that incorporates both associative and hierarchical structural features by passing dual-space messages through multilayer GNNs.
- To capture the dynamic effects of different modalities on various decision facts, we design an ISA module to fuse the cross-modal features in an interactive symmetric manner.
- Exhaustive experiments are conducted on two well-known MKG datasets. The effectiveness of DySarl in MKG reasoning is evident from the improvements achieved over all the baseline models across all the performance metrics.

The remainder of this paper is structured as follows. Section 2 discusses the related work. Section 3 details the DySarl model. The experimental analyses are described in Section 4, and the conclusions are presented in Section 5.

## 2 RELATED WORK

The existing modeling strategies for reasoning over MKGs can be divided into two categories: unimodal methods and multimodal methods. We make a summary about the categories of the existing MKG reasoning methods in Table 1 according to the ability of capturing complex multimodal structures.

### 2.1 Unimodal Reasoning Methods

Unimodal methods only consider the structural information contained in MKGs, excluding the visual and linguistic modalities. Early approaches can merely capture single-hop features. Among them, translation-based models such as TransE [3] and RotatE [24] minimize the relational distances between subjects and objects in valid factual triples. Tensor factorization-based methods such as ComplEx [26] and DistMult [35] establish connections between subjects

**Figure 2: The framework of DySarl. For a multimodal factual query (*Joe Biden*, *LiveAt*, *?*), the (a) DMS module learns the multihop structural features among the multimodal entities. The (b) AMI module extracts entity features from the multimodal auxiliary information. The (c) ISA module fuses all the modal features in an interactive and symmetric manner.**

and objects by decomposing the relational matrices. Convolution-based methods such as ConvE [8] and ConvTransE [22] and hyperbolic embedding-based method ATTH [5] achieve advanced performance in single-hop structural learning by modeling the one-hop associations and hierarchies in the Euclidean and hyperbolic spaces, respectively. Multihop path learning methods primarily rely on reinforcement learning to heuristically generate query-relevant reasoning paths; these methods include GaussianPath [31], DeepPath [34], MINERVA [7], and RLH [30]. Additionally, other multihop path learning methods, such as the transfer learning-based FIRE [39] and the rule-based NeuralLP [36], are available. GNN-based methods have been widely applied to multihop structural learning in unimodal KGs. Among them, association-based methods focus on capturing the multihop relational interactions between center entities with their Euclidean neighborhoods; these methods include CompGCN [28], RGCN [20], StarE [10], KBGAT [19], and so on. In contrast, hierarchy-based methods such as ReTIN [13] focus on perceiving multihop stratified features across neighborhoods through hyperbolic embedding-based GNNs. However, on the one hand, the existing hierarchy-based and association-based message-passing paradigms located in different spaces are separated and not yet unified, thus making it difficult to learn complete multihop structural features.On the other hand, the aforementioned methods fail to effectively integrate the multimodal information (e.g., texts and images) that is present in MKGs.

## 2.2 Multimodal Reasoning Methods

Recently, several MKG reasoning methods adapted for multimodal scenarios have been proposed. Focusing on the single-hop structures between multimodal entities in MKGs, some works utilize traditional concatenation or self-attention operations to extract entity-level auxiliary modal features and employ TransE to predict missing entities; these methods include TransAE [32] and IKRL [33]. In addition, MKRL [25] and KR-AMD [41] enhance entity representations using textual descriptions. Furthermore, MTRL [21] comprehensively integrates information derived from the structural, visual, and linguistic modalities, combining these multimodal features via the summation of subenergy functions. Optimal transport-based method OTKGE [4] focuses on aligning the distributions between low-dimensional vector embeddings of different modalities; this approach is a kind of coattention method as it emphasizes the interplay between modalities. However, the use of single-hop methods (e.g., TransE) to encode structural modalities in MKGs brings significant limitations to the aforementioned approaches. Among them, both MTRL and OTKGE have achieved relative state-of-the-art performance in single-hop MKG reasoning. MMKGR [40] is currently the state-of-the-art multihop path learning method for MKG reasoning; this model utilizes reinforcement learning to generate query paths for multimodal entities based on destinations, distances, and diverse rewards and fuses cross-modal features by

assigning coattention from the auxiliary modalities to the structural modality. However, on the one hand, the abovementioned approaches fail to capture complex structural features that lie beyond the multihop paths of the multimodal entities. On the other hand, the cross-modal feature fusion process in the previously developed methods is insufficient, overlooking the dynamic effects of different modalities on different reasoning facts. Hence, a unified framework for dynamically integrating the visual, linguistic, and multihop structural modalities in MKGs is needed.

## 3 METHODOLOGY

In this section, we present our proposed DySarl model. We first introduce the notations and definitions and provide a framework overview, which is followed by thorough explanations of its three modules. In addition, we elaborate on the training strategy and time complexity of DySarl.

### 3.1 Definitions and Notations

An MKG $\mathcal{G} = \{\mathcal{E}, \mathcal{V}, \mathcal{T}, \mathcal{R}, \mathcal{U}\}$ is an extension of a traditional KG where each entity is augmented with visual and linguistic data in addition to the structural modality. We define the entity, image, and text sets as $\mathcal{E}$, $\mathcal{V}$, and $\mathcal{T}$, each of which has a size of $N$. The relation set is defined as $\mathcal{R}$ with a size of $R$. $\mathcal{U} = \{(s, r, o)|s, o \in \mathcal{E}, r \in \mathcal{R}\}$ indicates a set of structural triples in the MKG $\mathcal{G}$, where $s$ and $o$ represent the subject and object entities and $r$ denotes the relation between them. For the entities $\{s, o\}$ and relation $r$ of a certain fact, their embeddings in the Euclidean space are represented as $s$, $o$, and $r$, respectively. The embedding dimensionality is set as $d$. We represent the Euclidean space and hyperbolic space as $\mathbb{R}$ and $\mathbb{B}$, respectively. The multimodal entity embedding matrices derived from the structures, images, and texts are defined as $E_S$, $E_I$, and $E_T$, respectively. In addition, we define the unimodal initialized entity embedding matrix as $E_U$. Finally, the multimodal unified entity embedding matrix is denoted as $E$, and the relation embedding matrix is denoted as $R$. In general, MKG reasoning aims to predict a missing subject $(?, r, o)$ or a missing object $(s, r, ?)$, where $r$ is a query relation and $\{s, o\}$ are multimodal entities equipped with structural, visual, and linguistic data. In practice, we unify these two tasks as object reasoning by adding inverse-relation factual triples $(o, r^{-1}, s)$ to $\mathcal{G}$.

### 3.2 Framework Overview

Figure 2 presents the overview framework of our proposed DySarl model. It consists of three modules: a dual-space multihop structural learning (DMS) module, an auxiliary modal information learning (AMI) module, and an interactive symmetric attention fusion (ISA) module. Specifically, the DMS module simultaneously incorporates both association-based and hierarchy-based features from the structural modality via the proposed dual-space message passing mechanism. A multilayer GNN architecture can learn the multihop structural information contained in the neighborhoods of MKGs. The AMI module extracts corresponding visual and linguistic features for each entity in an MKG using pretrained models. Finally, the ISA module employs symmetric attention to merge all the obtained multimodal features as attention targets while utilizing the initialized unimodal features as attention senders, thereby

equally assigning dynamic weights to different modal features. Furthermore, the fact-specific gated attention unit dynamically models the learnable weights of the multimodal unified attention targets and unimodal attention senders based on different decision facts during the reasoning stage.

### 3.3 Dual-Space Multihop Structural Learning

This module is responsible for learning the multihop structural features of a given MKG in terms of the associations and hierarchies observed in both the Euclidean and hyperbolic spaces.

As indicated in Figure 2(a), for each center entity $o$ to be aggregated, we represent its relation-specific neighborhoods as $\mathcal{E}_o^r$, which contain all entities $\{s\}$ connected to $o$ through relation $r$. Then, we employ relational messages [15, 20, 28] to aggregate the associative features that are directly linked to $o$ in the Euclidean neighborhoods. This message can be represented as follows:

$$\mathbf{msg}_e^{s,r} = s + r \tag{1}$$

where $\mathbf{msg}_e^{s,r}$, $s$, and $r \in \mathbb{R}^d$. Inspired by the superiority of hyperbolic embedding-based GNN approaches [13, 16] in terms of modeling multilevel (multihop) stratified data through a message aggregation process, we propose to design a hyperbolic message to perceive the hierarchical features contained across the neighborhoods of $o$. Formally, this message can be formulated in the following manner:

$$\mathbf{msg}_h^{s,r} = \log_{c_r}(\mathrm{H}_r(\exp_{c_r}(s)) \oplus^{c_r} \exp_{c_r}(r)) \tag{2}$$

where $\mathbf{msg}_h^{s,r} \in \mathbb{R}^d$. $\exp_{c_r}(\cdot)$ and $\log_{c_r}(\cdot)$ denote the exponential mapping and logarithmic mapping operations [5], which project a specific entity embedding (point) from the Euclidean space ($\mathbb{R}$) to the hyperbolic space ($\mathbb{B}$) and from the hyperbolic space ($\mathbb{B}$) back to the Euclidean space ($\mathbb{R}$), respectively. $c_r$ denotes the learnable relation-specific curvatures. $\oplus^{c_r}$ indicates the Möbius addition operation [27]. $\mathrm{H}_r(\cdot)$ aims to capture hierarchies in the hyperbolic space by learning isometric rotation and reflection operations for the entities $\{s\}$ in the neighborhoods of the aggregated entity $o$:

$$\mathrm{H}_r(s^{\mathbb{B}}) = Att(\mathrm{Rot}(\Theta_r)s^{\mathbb{B}}, \mathrm{Ref}(\Phi_r)s^{\mathbb{B}}; \mathbf{a}_r) \tag{3}$$

where $\mathrm{H}_r(s^{\mathbb{B}}) \in \mathbb{B}^d$. $s^{\mathbb{B}} = \exp_{c_r}(s) \in \mathbb{B}^d$ is the embedding of entity $s$ in hyperbolic space. $\mathrm{Ref}(\Phi_r)$ and $\mathrm{Rot}(\Theta_r)$ are hyperbolic isometries representing reflection and rotation operations [5], respectively, where $\Phi_r$ and $\Theta_r$ are relation-specific parameters. $Att(\cdot)$ learns an appropriate combination of hyperbolic reflections and rotations through a learnable attention weight $\mathbf{a}_r$.

Then, we design a message attention function to integrate Euclidean and hyperbolic messages. This process is formulated as:

$$\mathbf{msg}_{e\_h}^{s,r} = f(\omega^{\mathrm{T}}\mathbf{msg}_e^{s,r})\mathbf{msg}_e^{s,r} + f(\omega^{\mathrm{T}}\mathbf{msg}_h^{s,r})\mathbf{msg}_h^{s,r} \tag{4}$$

where $\mathbf{msg}_{e\_h}^{s,r} \in \mathbb{R}^d$ is the obtained dual-space message. $\omega$ is an attention vector, as presented in Figure 2(a). $f(\cdot)$ is the Softmax activation function. Then, we employ a multilayer GNN framework to aggregate both the multihop associative and hierarchical structural features among the multimodal entities of the given MKG. This process can be represented as follows:

$$o^{l+1} = \sigma\left(\sum_{r \in \mathcal{R}} \sum_{s \in \mathcal{E}_o^r} \mathrm{W}_r^l \mathbf{msg}_{e\_h}^{s,r,l} + \mathrm{W}_0^l o^l\right) \tag{5}$$

where $o^{l+1}$ and $o^l \in \mathbb{R}^d$ denote the embeddings of the aggregated entity $o$ in the $(l+1)^{th}$ and $l^{th}$ layers of the DMS module, respectively. $\mathbf{msg}_{e\_h}^{s,r,l} \in \mathbb{R}^d$ is the $l^{th}$-layer dual-space message (see Equation 4). $\mathrm{W}_r^l$ and $\mathrm{W}_0^l$ represent the learnable parameters for the relation-specific and self-loop structural features, respectively. $\sigma(\cdot)$ indicates the rectified linear unit (ReLU) [11] activation function.

Finally, through Equation 5, we can obtain the embedding matrix $E_S$ containing unified multihop structural features of the MKG:

$$E_S = \mathrm{DMS}(g, E_U) \qquad (6)$$

where $E_S \in \mathbb{R}^{N \times d}$. $E_U \in \mathbb{R}^{N \times d}$ is the initialized entity embedding matrix. $g$ records the structure of the given MKG. $\mathrm{DMS}(\cdot)$ represents the dual-space multihop structural learning process.

### 3.4 Auxiliary Modal Information Learning

This module aims to extract the auxiliary visual and linguistic features of each entity contained in the given MKG.

Following previous works [4, 21, 40], we utilize pretrained models to acquire features from the entity-level images and texts. Specifically, as shown in Figure 2(b), we retrieve the $d_I$-dimensional vector from the final fully connected layer prior to performing Softmax activation operation through a pretrained visual geometry group (VGG) [6, 23] model. For a specific entity $o$ in $\mathcal{E}$, this process can be formulated as follows:

$$I_o = \mathrm{W}_I \| pooling(\mathrm{VGG}(\mathcal{I}_o)) \| \qquad (7)$$

where $I_o \in \mathbb{R}^d$. $\mathrm{W}_I \in \mathbb{R}^{d_I \times d}$ is a learnable parameter for mapping the dimensionality of extracted image features to $d$. $\mathcal{I}_o \in \mathcal{V}$ refers to the collection of all images attached to $o$. $pooling(\cdot)$ denotes the mean pooling operation. $\| \cdot \|$ indicates the L2 normalization operation. To initialize the linguistic features derived from the textual descriptions, we utilize a pretrained word2vec [17] framework to encode the texts as a $d_T$-dimensional embedding for each entity. For a specific entity $o$ in $\mathcal{E}$, this process is represented as follows:

$$T_o = \mathrm{W}_T \| pooling(\mathrm{word2vec}(\mathcal{T}_o)) \| \qquad (8)$$

where $T_o \in \mathbb{R}^d$. $\mathrm{W}_T \in \mathbb{R}^{d_T \times d}$ is a learnable parameter for mapping the dimensionality of the textual features to $d$. Additionally, $\mathcal{T}_o \in \mathcal{T}$ indicates the description words attached to $o$.

Finally, by stacking the image features $\{I_o\}$ and textual features $\{T_o\}$ of all entities, we can obtain entity embedding matrices $E_I \in \mathbb{R}^{N \times d}$ and $E_T \in \mathbb{R}^{N \times d}$ for the visual and linguistic modalities of the given MKG, respectively.

### 3.5 Interactive Symmetric Attention Fusion

This module aims to fuse all the modal features and capture the dynamic effects of different modalities on different decision facts.

Inspired by the superiority of attention architectures [13, 29, 40] in terms of integrating different elements, as shown in Figure 2(c), we design a symmetric attention component to obtain multimodal unified entity embeddings. We uniformly learn the obtained multimodal features $\{E_S, E_I, E_T\}$ as attention targets and assign different attention weights to them using the initialized unimodal features $E_U$. Specifically, $E_U$ is used to generate the query matrix for the multi-head attention (MHA) process, while $\{E_S, E_I, E_T\}$ is used to generate the key and value matrices:

$$E_{MHA} = \sigma\left(\frac{\mathrm{W}_q E_U (\mathrm{W}_k[E_S; E_I; E_T])^T}{\sqrt{d_k}}\right) \mathrm{W}_v[E_S; E_I; E_T] \quad (9)$$

where $E_{MHA} \in \mathbb{R}^{N \times d}$ denotes the temporary output embeddings. $[;]$ represents the concatenation operation. $d_k$ is a scaling factor for preventing vanishing gradients. $\mathrm{W}_q$, $\mathrm{W}_k$, and $\mathrm{W}_v$ are learnable parameters used to fit the weights of different modal features. Then, a feed-forward network (FFN) with $d$ hidden units is introduced:

$$E = \mathrm{W}_1(\sigma(\mathrm{W}_2 E_{MHA})) \qquad (10)$$

where $E \in \mathbb{R}^{N \times d}$ indicates the multimodal unified entity embeddings. $\mathrm{W}_1$ and $\mathrm{W}_2 \in \mathbb{R}^{d \times d}$ are learnable parameters that introduce more semantics. Note that we employ layer normalization [1] and residual connections [12] after the MHA and FFN stages.

Then, we design a fact-specific gated attention unit to learn a unique weight of the features derived from multimodal unified embeddings as attention targets and unimodal initialized embeddings as attention senders based on each specific decision fact during the reasoning phase. Following previous work [13], to perceive hierarchies during the decoding process, we use the ATTH [5] model as a decoder to generate scores for both the attention target and sender encoding features. Specifically, for a certain factual query $(s, r, ?)$, ATTH calculates the hyperbolic distances between all candidate entities and the Möbius summation of $s$ and $r$, assigning higher scores to entities that are closer in distance:

$$\mathcal{S}_1 = \nabla(\mathrm{ATTH}(s, r)E), \mathcal{S}_2 = \nabla(\mathrm{ATTH}(\hat{s}, r)E_U) \qquad (11)$$

where $s \in E$, $\hat{s} \in E_U$, and $r \in R$. $\mathcal{S}_1$ and $\mathcal{S}_2 \in \mathbb{R}^N$ are scores generated from the attention target and sender embedding matrices, respectively. $\nabla(\cdot)$ indicates the sigmoid activation function. Then, we introduce two learnable parameters $\{\delta_s, \delta_r\} \in \mathbb{R}^N$ to dynamically adjust the attention of a specific fact on different modal features during the reasoning stage. This process is formulated as follows:

$$\mathcal{S} = \nabla(\delta)\mathcal{S}_1 + (1 - \nabla(\delta))\mathcal{S}_2 \qquad (12)$$

where $\delta = \mathrm{Max}(\delta_s[s], \delta_r[r]) \in \mathbb{R}$ denotes the learning weight for a specific query, where $\delta_s[\cdot]$ and $\delta_r[\cdot]$ represent the retrieval values of a specific dimension from the vectors. $\mathcal{S} \in \mathbb{R}^N$ is the final reasoning score, where each dimension is the probability of predicting the corresponding multimodal entity as the missing object.

### 3.6 Training Strategy

We adopt a multilabel learning framework to train the DySarl model through the cross-entropy loss function:

$$\mathcal{L} = \sum_{(s,r,o) \in \hat{\mathcal{U}}} \sum_{i=0}^{N-1} y_i \log(\mathcal{S}_i) \qquad (13)$$

where $\hat{\mathcal{U}} \subset \mathcal{U}$ contains the factual triples included in the training set. $y_i$ is 1 if the training fact completed by the $i^{th}$ entity is valid, and 0 otherwise. $\mathcal{S}_i$ denotes the score produced for the $i^{th}$ entity when reasoning about the current training fact.

### 3.7 Computational Complexity Analysis

To demonstrate the time efficiency of the DySarl model, we analyze the computational complexities of its three modules. The DMS module uses a GNN-based framework to perform dual-space message passing with a depth of $L$ layers, resulting in a time complexity

**Table 2: MKG reasoning performance (in percentages) achieved across the WN9-IMG-TXT and FB-IMG-TXT datasets in terms of raw metrics. The best and second-best results are bolded and underlined, respectively.**

| Model | WN9-IMG-TXT | | | | FB-IMG-TXT | | | |
|---|---|---|---|---|---|---|---|---|
| | MRR | H@1 | H@5 | H@10 | MRR | H@1 | H@5 | H@10 |
| ConvTransE | 59.96 | 50.83 | 70.74 | 77.41 | 28.97 | 17.34 | 41.96 | 53.74 |
| ATTH | 59.44 | 50.53 | 70.36 | 76.88 | 26.70 | 14.92 | 40.08 | 52.44 |
| FIRE | 56.40 | 52.80 | 77.60 | 86.80 | 42.80 | 37.90 | 49.50 | 57.10 |
| RLH | 62.40 | 58.30 | 81.30 | 89.40 | 50.60 | 44.50 | 60.20 | 68.40 |
| RGCN | 65.04 | 48.79 | 84.34 | 88.44 | 76.97 | 65.06 | 91.69 | 95.04 |
| ReTIN | 74.18 | 65.28 | 85.22 | 90.98 | 88.82 | 82.36 | 97.07 | 98.86 |
| MTRL | 48.30 | 45.60 | 69.80 | 83.80 | 25.20 | 21.30 | 32.40 | 47.20 |
| OTKGE | 61.54 | 51.74 | 73.24 | 80.25 | 29.14 | 17.26 | 42.57 | 55.13 |
| MMKGR | 80.20 | 73.60 | 87.80 | 92.80 | 71.30 | 65.80 | 77.50 | 82.60 |
| **DySarl** | **88.65** | **83.70** | **94.69** | **96.40** | **93.72** | **89.62** | **98.85** | **99.84** |
| ΔImprove. | 10.5% | 13.7% | 7.85% | 3.88% | 5.52% | 8.81% | 1.83% | 1.00% |

of $O(LdN)$. For the AMI module, the time complexity for extracting visual features is $O(NmI\mathcal{A}k^2c^2)$, where $\mathcal{A}$ represents the image size; $I$, $k$, and $c$ denote the depth, kernel size, and maximum number of channels of the VGG model; and $m$ represents the number of images attached to each entity. The time complexity for extracting linguistic features is $O(Nn)$, where $n$ is the description length attached to each entity. For the ISA module, the time complexity of the symmetric attention component is $O(\mathcal{P}^2 d)$, where $\mathcal{P}$ represents the number of modalities. The time complexity of the decoder and the fact-specific gated-attention unit is $O(|\mathcal{U}|dN)$.

## 4 EXPERIMENTS

### 4.1 Experimental Setup

*4.1.1 Datasets.* We assess our proposed DySarl model on two public MKG datasets[1]: FB-IMG-TXT [21] and WN9-IMG-TXT [33]. They are both widely adopted by the existing MKG reasoning methods [4, 21, 40], each entity of which contains an item from each of three modalities: structures, texts, and images. Specifically, the structural factual triples and textual descriptions of the FB-IMG-TXT and WN9-IMG-TXT datasets are extracted from Freebase [2] and Word-Net [18], respectively. To supplement the visual modality, 100 and 10 images are crawled for each entity in FB-IMG-TXT and WN9-IMG-TXT, respectively. The statistics are detailed in Table 3.

*4.1.2 Baseline Models.* We compare our proposed DySarl model with multiple representative unimodal and multimodal methods for MKG reasoning. Specifically, among the unimodal approaches, the single-hop comparison methods include ConvTransE [22] and ATTH [5]; the multihop path learning models include FIRE [39] and RLH [30]; and the multihop structural learning methods include RGCN [20] and ReTIN [13]. For the multimodal approaches, we conduct comparisons with the representative single-hop models, including MTRL [21] and OTKGE [4], and the state-of-the-art multihop path learning method MMKGR [40]. The baseline methods are described in detail in Section 2.

---

[1]https://public.ukp.informatik.tu-darmstadt.de/starsem18-multimodalKB

**Table 3: Statistical information of the utilized MKG datasets.**

| Datasets | #Training | #Validation | #Test | #Ent | #Rel | #Scale |
|---|---|---|---|---|---|---|
| WN9-IMG-TXT | 11,747 | 1,337 | 1,319 | 6,555 | 9 | Small |
| FB-IMG-TXT | 285,850 | 29,580 | 34,863 | 11,757 | 1,231 | Large |

*4.1.3 Evaluation Metrics.* We use the widely adopted mean reciprocal ranking (MRR) and Hits@K (H@K) metrics to evaluate the tested models. MRR reflects the mean reasoning ranking of ground-truth entities across all query facts. H@K indicates the proportion of facts in the top K prediction hits out of the total number of query facts. Following previous work [40], we choose $K \in \{1, 5, 10\}$ and report the mean results of multimodal object and subject reasoning tasks. Finally, to reflect the original capabilities of the tested models, we adopt the raw setting without filtering operations.

*4.1.4 Implementation Details.* We implement and train our DySarl model using PyTorch on a single RTX A5000 GPU. We configure the parameters according to the model performance on the validation set in terms of the MRR metric. The batch size is set as 1000. We set the training epochs to 200 to guarantee model convergence. The embedding dimensionality $d$ is set to 100. For the DMS module, we set the number of GNN layers for dual-space multihop message passing to 2. For the AMI module, we use the VGG19 and VGG-m-128 models to produce 4096- and 128-dimensional (corresponding to $d_I$) embeddings for each image in the WN9-IMG-TXT and FB-IMG-TXT datasets, respectively. Additionally, the dimensionalities $d_T$ of each text generated for the FB-IMG-TXT and WN9-IMG-TXT datasets are 1000 and 300, respectively. For the ISA module, the number of symmetric attention heads is set to 2. We use the Adam optimizer for training and the learning rate is set to 0.001.

Following the same datasets and evaluation settings, some of the baseline results are adopted from [40]. We replicate the results of the ConvTransE, ATTH, RGCN, ReTIN, and OTKGE models under the same experimental settings. Note that ConvTransE and ATTH are employed as the decoders of the RGCN and ReTIN, respectively. For the MMKGR model without open-source codes, we adopt the best results reported in its original paper.

### 4.2 Reasoning Results Obtained over MKGs

In this section, we compare the proposed DySarl model with multiple representative unimodal and multimodal baseline methods. The main MKG reasoning results are presented in Table 2.

All the multimodal methods except MTRL exhibit significantly superior performance to that of the unimodal single-hop methods represented by ConvTransE and ATTH. This disparity can be attributed to the utilization of TransE by MTRL for capturing single-hop features in a limited manner, as well as its coarse concatenation-based multimodal fusion step. The unimodal multihop path learning methods, represented by FIRE and RLH, and the unimodal multihop structural learning methods, represented by RGCN and ReTIN, outperform the multimodal single-hop methods, including MTRL and OTKGE. This finding suggests that the graph-level complex structural features among multimodal entities can effectively yield improved MKG reasoning performance. The multimodal multihop path learning method, MMKGR, achieves significant performance

**Table 4: Ablation results achieved across all the datasets.**

| Datasets | WN9-IMG-TXT | FB-IMG-TXT |
|---|---|---|
| w/o Multihop Associative Structural Features | 70.98 | 88.86 |
| w/o Multihop Hierarchical Structural Features | 62.22 | 14.05 |
| w/o Visual Features | 80.31 | 92.95 |
| w/o Linguistic Features | 85.48 | 93.03 |
| w/o Attention Sender Features | 79.51 | 89.99 |
| w/o Attention Target Features | 59.11 | 26.70 |
| **DySarl** | **88.65** | **93.72** |

on the WN9-IMG-TXT dataset due to its ability to capture query-relevant path information and its incorporation of coattention for fusing multimodal features. However, it performs worse than the RGCN and ReTIN on the FB-IMG-TXT dataset. This indicates that the multihop structural learning methods can perform better on large-scale graphs, such as the FB-IMG-TXT dataset.

The proposed DySarl model outperforms all the unimodal and multimodal reasoning methods across all the MKG datasets and evaluation metrics. This can be attributed to two key factors. First, DySarl comprehensively integrates the auxiliary visual and linguistic features derived from the entire input MKG in addition to the structural modality. Second, DySarl is capable of capturing complete complex features that lie beyond the paths contained in the structural modality. Moreover, in the fusion step, DySarl dynamically highlights the influences of different modal features. We observe that the improvement achieved by DySarl over ReTIN is relatively small in terms of the H@5 and H@10 metrics produced on the FB-IMG-TXT dataset. This highlights the fact that multihop structural learning methods, especially hierarchy-based approaches, can roughly overcome the loss caused by the lack of multimodal auxiliary information in large-scale graphs (as reflected by the H@5 and H@10 metrics). However, due to the ability to effectively integrate both association-based and hierarchy-based structural features, as well as multimodal features, DySarl achieves significant improvements over the state-of-the-art baseline models in terms of overall reasoning performance and accurate predictions (as reflected by the MRR and H@1 metrics).

### 4.3 Ablation Study

In this section, we investigate the impact of each variant of DySarl on its MKG reasoning performance using the MRR metric.

As presented in Table 4, the DySarl (w/o Multihop Associative Structural Features) and DySarl (w/o Multihop Hierarchical Structural Features) variants remove the Euclidean and hyperbolic messages, respectively, from the multihop structural aggregation process of the DMS module. The DySarl (w/o Visual Features) and DySarl (w/o Linguistic Features) variants exclude the entity embeddings derived from images and texts in the AMI module, respectively. The DySarl (w/o Attention Sender Features) variant applies coattention derived from the structural features to all modal features in the ISA module. The DySarl (w/o Attention Target Features) variant solely utilizes the initialized embeddings as attention senders to generate the final reasoning scores in the ISA module.

The results indicate that multihop structural features contribute more to MKG reasoning than do auxiliary visual and linguistic

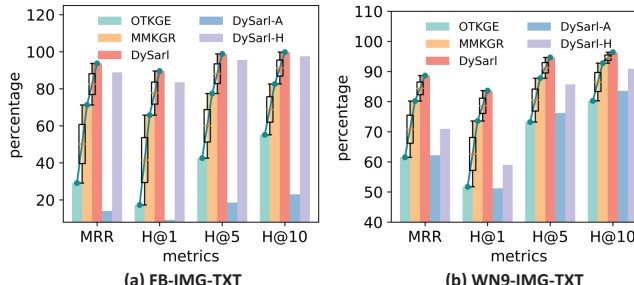

**Figure 3: Study on the multihop structural features over MKGs on the FB-IMG-TXT and WN9-IMG-TXT datasets.**

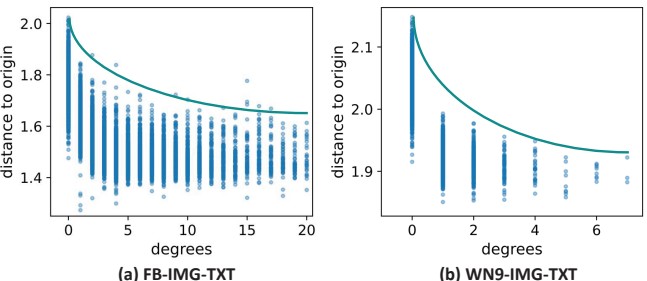

**Figure 4: Study on the hierarchical features over MKGs on the FB-IMG-TXT and WN9-IMG-TXT datasets.**

modal features. DySarl (w/o Multihop Hierarchical Features) is significantly weaker than DySarl (w/o Multihop Associative Features), especially on the FB-IMG-TXT dataset. On the one hand, this suggests that the role of the hierarchical features conveyed by hyperbolic message passing is more significant than that of the associative features conveyed by Euclidean message passing. On the other hand, the FB-IMG-TXT MKG contains more distinct multihop hierarchical structures. The performance of DySarl (w/o Visual Features) is relatively inferior to that of DySarl (w/o Linguistic Features). This indicates that within the auxiliary data of MKGs, images play a more prominent role in improving the reasoning capabilities, surpassing the contributions of texts. During the cross-modal feature fusion stage, DySarl outperforms DySarl (w/o Attention Sender Features), demonstrating the superiority of the symmetric attention component incorporated in the ISA module over the conventional asymmetric coattention mechanism. Moreover, DySarl (w/o Attention Target Features) exhibits a significant decrease in performance compared to that of DySarl. This is due to the fusion of all modal features as attention targets for obtaining the unified multimodal embeddings; however, the unimodal initialized embeddings contain extremely limited patterns. Ultimately, the ability of DySarl to outperform all its variants proves that each component contributes to the overall performance of the model.

### 4.4 On the Multihop Structures

In this section, we investigate the impact of the multihop structural features among multimodal entities on MKG reasoning.

As shown in Figure 3, OTKGE and MMKGR are representative multimodal one-hop learning and multihop path learning methods, respectively. The DySarl-A and DySarl-H variants selectively pass Euclidean and hyperbolic messages to consider association-based

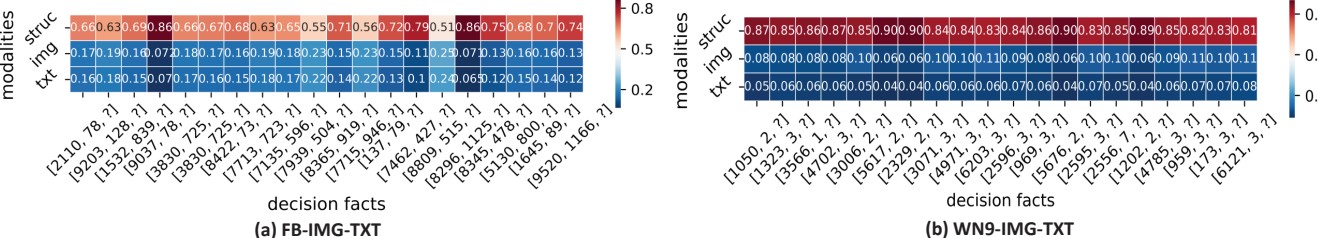

**(a) FB-IMG-TXT**                                                        **(b) WN9-IMG-TXT**

**Figure 5: Study on the cross-modal attentive dynamics in MKG reasoning on the FB-IMG-TXT and WN9-IMG-TXT datasets.**

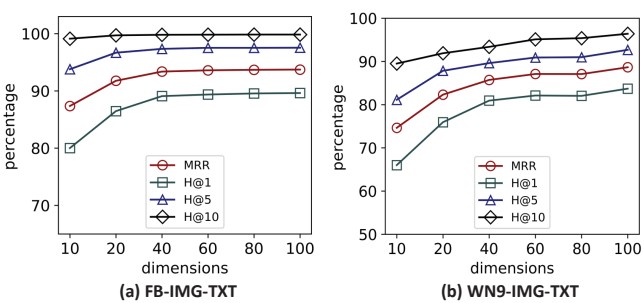

**(a) FB-IMG-TXT**                              **(b) WN9-IMG-TXT**

**Figure 6: Study on the low-dimensional stability of DySarl on the FB-IMG-TXT and WN9-IMG-TXT datasets.**

and hierarchy-based multihop structural features, respectively. It can be observed that the method based on multihop paths can capture more structural features than the single-hop method. Due to its ability to model the complex structures that lie beyond paths, the proposed multihop structural method DySarl achieves further performance improvements. This process of improvements at the structural level is more pronounced on large-scale graphs (e.g., FB-IMG-TXT), which aligns with the results in Section 4.3. In addition, the multihop structural features of hierarchies exert stronger influences than those of associations during MKG reasoning.

We further demonstrate that DySarl has the ability to capture the multilevel hierarchical features among multimodal entities. As presented in Figure 4, we plot the relationships and trends between the degrees of the entities and their hyperbolic distances to the origin in the reasoning stage. The points with higher degrees (i.e., active entities) are at a higher level and closer to the origin. Despite the additional noise introduced by the presence of auxiliary modal information, DySarl effectively learns distinct multilevel (multihop) hierarchical features for the multimodal entities in MKGs. Furthermore, the FB-IMG-TXT MKG exhibits a more pronounced hierarchical structure, which aligns with the findings in Section 4.3.

### 4.5 On the Attentive Dynamics

In this section, we demonstrate that the multimodal feature fusion process of DySarl can capture the dynamic effects of different modal features on various decision facts in the reasoning stage.

As illustrated by the x-axes in Figure 5, we randomly extract 20 decision fact samples from the reasoning graphs of the FB-IMG-TXT and WN9-IMG-TXT MKGs during the testing phase. The y-axes represent the three different modalities: structural, visual, and linguistic. Subsequently, we depict the dynamic attention of the sampling facts on the three modalities contained within the MKGs during the inference process. Specifically, we utilize the feature

embeddings derived from different modalities to generate prediction scores for each specific decision fact. Notably, the fact-specific gated attention unit combines the scores obtained from the attention senders for each individual modality (serving as the attention target). Then, we perform L1 normalization on a set of different modal scores [$struc$, $img$, $txt$] for each prediction fact.

It can be observed that the structural modality plays the most crucial reasoning role for both MKGs. Compared to the WN9-IMG-TXT dataset, the FB-IMG-TXT dataset benefits more from the visual and linguistic modalities, which can be attributed to the presence of more additional auxiliary images and words that provide more effective multimodal representations. On the one hand, from a horizontal perspective, each individual modality contributes distinctively to the predictions of different facts. On the other hand, from a vertical perspective, specific decision facts exhibit varying levels of attention towards different modal features. Hence, our proposed DySarl model effectively captures the attentive dynamics of the cross-modal fusion process.

### 4.6 On the Embedding Dimensions

Given the compatibility of hyperbolic embeddings [5, 13] with low dimensions, in this section, we study the influences of different embedding dimensions on the performance of the DySarl model. As illustrated in Figure 6, we conduct experiments under varying dimensional settings: $d \in \{10, 20, 40, 60, 80, 100\}$. It can be observed that as the number of embedding dimensions increases, the model performance improves on both datasets. This is because higher dimensional embeddings can store more valuable features. Furthermore, DySarl maintains a certain degree of performance stability under low dimensions, especially for the FB-IMG-TXT MKG. This is because the superlinear growth property of hyperbolic distance enables the effective expression of complex structures among data using limited dimensions, particularly hierarchical structures.

### 5 CONCLUSIONS

In this paper, we propose DySarl to address the challenges of graph-level multimodal multihop structures and entity-level cross-modal attentive dynamics encountered in MKG reasoning. Specifically, DySarl excels in capturing complex multihop structures beyond limited reasoning paths during the process of structural modality learning. Moreover, DySarl can effectively highlight the dynamic influence of different modalities on different reasoning facts during the cross-modal feature fusion process. By learning accurate representations of multimodal entities in MKGs, our experiments demonstrate the significant reasoning performance improvements of DySarl over the baseline models.

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
