# OpenReview forum: "DySarl: Dynamic Structure-Aware Representation Learning for Multimodal Knowledge Graph Reasoning"
_acmmm.org/ACMMM/2024/Conference — MM2024 Poster_

### Official Review · Reviewer_hRE4 · 2024-05-17

**Rating:** 5
**Confidence:** 3

**Summary:**

The main work of this paper is to propose a new model called DySarl for reasoning in multimodal knowledge graphs (MKG). DySarl aims to address the limitations of existing methods in learning complex structural modalities and cross-modal attention dynamics.
DySarl designed a dual-space multi-hop structure learning (DMS) module to effectively aggregate multi-modality through a new message passing mechanism while considering the correlation and hierarchical structure features in Euclidean space and hyperbolic space. Multi-hop structural characteristics of entities. This helps capture complex neighborhood information beyond the inference path.
DySarl also proposed the Interactive Symmetric Attention Fusion (ISA) module: This module dynamically learns the impact of different modal features on decision-making facts through a newly designed symmetric attention component and fact-specific gated attention unit. This enables DySarl to dynamically balance the contribution of different modal features in the inference process.
Moreover, experiments on two public MKG datasets show that DySarl significantly outperforms existing single-modal and multi-modal baseline models on multi-modal reasoning tasks, especially in capturing multi-modal multi-hop structures and Aspects of dynamic cross-modal attention.
In general, DySarl improves the accuracy and efficiency of MKG reasoning by dynamically integrating multi-modal information and structural features, and provides new perspectives and methods for multi-modal knowledge graph reasoning.

**Strengths:**

Novelty: DySarl proposes a new dynamic structure-aware representation learning framework that combines dual-space multi-hop structure learning and interactive symmetric attention fusion, which is unique among existing methods. It not only considers complex structural interactions between multi-modal entities, but also dynamically fuses features of different modalities, thereby improving inference performance.

Theoretical and technical correctness: DySarl utilizes the properties of Euclidean and hyperbolic spaces to capture association and hierarchical structure features, which is innovative in the representation learning of multi-modal knowledge graphs. At the same time, the model uses a multi-head attention mechanism and a gating unit to ensure dynamic weight distribution of different modal features during the reasoning process.

Adequate evaluation: The paper conducted experiments on two public multi-modal knowledge graph data sets and compared it with a variety of baseline models. The results show that DySarl has achieved significant performance improvements on multiple evaluation indicators, proving the effectiveness of its methods.

Clarity: Although there are some details that may require further clarification, the paper's overall description of the model's architecture, working principle, and experimental setup is detailed, helping readers understand its core ideas.

Application potential: DySarl’s multi-modal reasoning capabilities are suitable for a variety of real-world application scenarios, such as information retrieval, social media analysis, and intelligent question-answering systems, especially in complex tasks that require the combination of structural, visual, and linguistic information. In addition, the paper provides an open-source implementation of the model, which promotes the reproduction and further improvement of the model by other researchers and developers.

**Limitations:**

Lack of novelty:  DySarl is very excellent in terms of innovation. But in terms of model clarity, understanding how the model dynamically assigns weights to different decision facts and learns complex structural features may require further explanation. In addition, regarding the scalability of the model, the paper does not discuss how the model adapts to the growing MKG, or how to handle the dynamic addition of new entities and relationships. This may be a potential limitation that needs to be considered in future extension work.

Technical errors: In the specific method introduction in Chapter 3, regarding the details of the DMS module, the details in Chapter 3.3 of how to combine message passing in Euclidean and hyperbolic spaces to learn association and hierarchical features simultaneously may not be detailed enough. A deeper mathematical description or pseudocode may be needed to aid understanding. In addition, regarding the dynamic weights of the ISA module, the details of how to learn the dynamic weights to adapt to different decision facts may not be clear enough. This may require a more detailed explanation of how attention weights are dynamically adjusted during inference.

Insufficient evaluation: In Section 4.1, about the selection of data sets. The paper only evaluates on two public datasets, which may not fully reflect the model's performance on wider or more complex MKGs. It is suggested that using more or larger datasets for validation would increase confidence in the model's ability to generalize. In the experimental settings in Chapter 4.1, there are still some questions about the specific settings of training and verification. For example, except for the learning rate adjustment and hyperparameter selection of the ISA module, the experimental setting data of other modules are not given. This information is critical for other researchers to replicate the results.

**Suitability:**

2

---

### Official Review · Reviewer_niQT · 2024-05-27

**Rating:** 6
**Confidence:** 3

**Summary:**

This paper proposes a novel Dynamic Structure-aware Representation Learning method, namely DySarl, to overcome this problem and significantly improve MKG reasoning performance. Specifically, the authors devise a dual-space multihop structural learning module in DySarl, which aggregates the multihop structural features of multimodal entities via a novel message-passing mechanism. This mechanism integrates message paradigms in both Euclidean and hyperbolic spaces, effectively preserving neighborhood information beyond the limited multimodal query paths.

Furthermore, DySarl incorporates an interactive symmetric attention module to explicitly learn the dynamic impacts of unimodal attention senders and multimodal attention targets on decision facts. This is achieved through a newly designed symmetric attention component and a fact-specific gated attention unit. These features equip DySarl with dynamic associations between multimodal feature learning and subsequent reasoning.

Extensive experiments demonstrate that DySarl outperforms state-of-the-art baselines in reasoning performance on two public MKG datasets.

**Strengths:**

This paper introduces DySarl, a pioneering Dynamic Structure-aware representation learning method aimed at addressing this challenge and notably enhancing MKG reasoning performance.

Extensive experiments demonstrate that DySarl delivers substantial improvements in reasoning performance across two public MKG datasets compared to state-of-the-art baselines.

**Limitations:**

The technical details are somewhat complex, and the motivation behind the research is not clearly articulated. Additionally, many symbols lack clear definitions.

Furthermore, the experimental results lack in-depth analysis.

Missing references:

 Is Visual Context Really Helpful for Knowledge Graph? A Representation Learning Perspective

Hybrid Transformer with Multi-level Fusion for Multimodal Knowledge Graph Completion

**Suitability:**

2

---

### Official Review · Reviewer_Pk4j · 2024-05-28

**Rating:** 3
**Confidence:** 3

**Summary:**

The paper focuses on multimodal knowledge graphs reasoning and the authors argue that existing methods struggle with capturing complex structural interactions and the dynamic impact of multimodal features on reasoning. To address this, they propose a method named DySarl, which incorporates a dual-space multi-hop structural learning module and an interactive symmetric attention module.  Experiments on two public MKG datasets (FB-IMG-TXT and WN9-IMG-TXT) demonstrate their method's effectiveness.

**Strengths:**

1. The paper is well-organized and easy to follow.
2. Their method obtain noticeable performance on two multimodal knowledge graph reasoning datasets FB-IMG-TXT and WN9-IMG-TXT.
3. The authors conduct a detailed ablation study and in-depth analysis analysis to some extent.

**Limitations:**

1. The idea that embedding entity structure of knowledge graphs in Euclidean and hyperbolic spaces is not novel. Although the whole method DySarl is complex and complete, each component of the method is unremarkable.

2. The author only conducted experiments on two datasets, FB-IMG-TXT and WN9-IMG-TXT. Perhaps comparisons could be made with more advanced methods on more datasets, such as FB15k237-IMG and WN18-IMG.

3. Does the author consider trying to more powerful visual and text encoders? For example, the multimodal LLM such as LLAVA and others.

**Suitability:**

2

---

### Meta-Review · Area_Chair_P7y7 · 2024-06-28

**Recommendation:** Accept (Poster)
**Confidence:** 5

**Metareview:**

The paper proposes a Dynamic Structure-aware Representation Learning method, DySarl, for reasoning in multimodal knowledge graphs. DySarl integrates a dual-space multi-hop structural learning module and an interactive symmetric attention module to address the complexities of multimodal features and their dynamic interactions. Experiments on two public MKG datasets demonstrate improvements over state-of-the-art baselines. Strengths include the novel approach, robust theoretical foundation, and thorough experimental validation. However, the reviewers noted concerns about the complexity of the method, the need for more comprehensive evaluations on additional datasets, and the lack of discussion on the scalability and efficiency of the model. The authors well addressed some concerns during the rebuttal. Given the paper's overall quality and significant contributions to the field, I recommend acceptance.